# Mathematical Modeling of Fluconazole Resistance in the Ergosterol Pathway of *Candida albicans*

Paul K. Yu,[a,b,d] Llewelyn S. Moron-Espiritu,[a,c] Angelyn R. Lao[a,d]

aSystems and Computational Biology Research Unit, Center for Natural Sciences and Environmental Research, De La Salle University, Malate, Manila, National Capital Region, Philippines
bMS Graduate Fellow - Career Incentive Program, Science Education Institute, Department of Science and Technology, Taguig City, National Capital Region, Philippines
cDepartment of Biology, De La Salle University, Malate, Manila, National Capital Region, Philippines
dDepartment of Mathematics and Statistics, De La Salle University, Malate, Manila, National Capital Region, Philippines

**ABSTRACT** Candidiasis is reported to be the most common fungal infection in the critical care setting. The causative agent of this infection is a commensal pathogen belonging to the genus *Candida*, the most common species of which is *Candida albicans*. The ergosterol pathway in yeast is a common target by many antifungal agents, as ergosterol is an essential component of the cell membrane. The current antifungal agent of choice for the treatment of candidiasis is fluconazole, which is classified under the azole antifungals. In recent years, the significant increase of fluconazole-resistant *C. albicans* in clinical samples has revealed the need for a search for other possible drug targets. In this study, we constructed a mathematical model of the ergosterol pathway of *C. albicans* using ordinary differential equations with mass action kinetics. From the model simulations, we found the following results: (i) a partial inhibition of the sterol-methyltransferase enzyme yields a fair amount of fluconazole resistance; (ii) the overexpression of the *ERG6* gene, which leads to an increased sterol-methyltransferase enzyme, is a good target of antifungals as an adjunct to fluconazole; (iii) a partial inhibition of lanosterol yields a fair amount of fluconazole resistance; (iv) the C5-desaturase enzyme is not a good target of antifungals as an adjunct to fluconazole; (v) the C14$\alpha$-demethylase enzyme is confirmed to be a good target of fluconazole; and (vi) the dose-dependent effect of fluconazole is confirmed. This study hopes to aid experimenters in narrowing down possible drug targets prior to costly and time-consuming experiments and serve as a cross-validation tool for experimental data.

**IMPORTANCE** Candidiasis is reported to be the most common fungal infection in the critical care setting, and it is caused by a commensal pathogen belonging to the genus *Candida*, the most common species of which is *Candida albicans*. The current antifungal agent of choice for the treatment of candidiasis is fluconazole, which is classified under the azole antifungals. There has been a significant increase in fluconazole-resistant *C. albicans* in recent years, which has revealed the need for a search for other possible drug targets. We constructed a mathematical model of the ergosterol pathway in *C. albicans* using ordinary differential equations with mass action kinetics. In our simulations, we found that by increasing the amount of the sterol-methyltransferase enzyme, *C. albicans* becomes more susceptible to fluconazole. This study hopes to aid experimenters in narrowing down the possible drug targets prior to costly and time-consuming experiments and to serve as a cross-validation tool for experimental data.

**KEYWORDS** *Candida albicans*, fluconazole resistance, ergosterol pathway, mathematical modeling, systems biology

Address correspondence to Angelyn R. Lao, angelyn.lao@dlsu.edu.ph.

The authors declare no conflict of interest.

Candidiasis refers to a broad range of fungal infections caused by different species of a fungal commensal pathogen belonging to the genus *Candida*. The following *Candida* species are all reported to cause cutaneous, mucosal, bloodstream, and deep-

**FIG 1** Chemical structure of fluconazole (adopted from [6]). It is a member of the triazole class and is a five-membered ring with 3 nitrogen atoms (shown on the left and right). It is also comprised of difluorobenzene, a benzene ring with 2 fluorine atoms (shown at the bottom).

seated tissue infections: *Candida albicans*, *C. glabrata*, *C. tropicalis*, *C. parapsilosis*, and *C. krusei*. The most common of these is *Candida albicans* (1).

Invasive candidiasis is the term used for the more concerning type, which includes bloodstream yeast infections, which are also known as candidemia. Likewise, deep-seated tissue infections are infections that occur in sterile sites, such as the peritoneal cavity, and in abdominal and nonabdominal sites, namely, the bones, muscles, joints, eyes, or central nervous system (2). Surprisingly, the invasive type is the most common fungal infection in the critical care setting, with crude mortality of about 40 to 55% (3).

Logan et al. (2020) noted several risk factors for invasive candidiasis that include the use of broad-spectrum antimicrobials, immunosuppressive drugs, total parenteral nutrition, iatrogenic interventions, along with several challenges in the critical care setting, namely, the shift to more resistant *Candida* epidemiology and the appropriate strategies for antifungal therapy (3). Among the different classes of antifungal agents used for invasive candidiasis are agents such as azoles, polyenes, echinocandins, and allylamines. However, fluconazole, a fungistatic drug belonging to the azole group, is currently the preferred treatment for *C. albicans* infections due to its low cost, limited toxicity, and oral administration (4). The mechanism of action of azole drugs is the inhibition of the C14$\alpha$-demethylase enzyme, which blocks the synthesis of ergosterol, an essential component of the fungal cell membrane in *Candida* species (5). The chemical structure of fluconazole is depicted in Fig. 1, as adopted from (6).

In recent years, Whaley et al. (2017) noted a high level of resistance to azoles among *Candida* species, and some nonalbican *Candida* species have already been reported to have intrinsic resistance to azoles (4). In fact, the Centers for Disease Control and Prevention (CDC), in its Antibiotic Resistance Threats in the United States, 2019 report, classifies drug-resistant *Candida* as a serious threat, with an estimate of 34,800 cases and 1,700 deaths (7). The CDC noted that about 7% of all *Candida* blood samples tested are resistant to fluconazole (7).

There is an abundance of experimental investigations into azole resistance mechanisms in *C. albicans*, which can be costly and time-consuming. In this study, we propose a different approach to investigating fluconazole resistance in *C. albicans*, namely, using mathematical models to determine possible drug targets within the ergosterol pathway. The aim of this study is to narrow down possible drug targets prior to costly and time-consuming experiments and to serve as a cross-validation tool for experimental data. To our knowledge, this is the first study to investigate the fluconazole resistance in the ergosterol pathway of *Candida albicans* using mathematical modeling.

The rest of the paper is organized as follows: The results of the model simulations are shown in the Results section. These are followed by a discussion on how the results compare with previous literature, the conclusions drawn, and further recommendations that are proposed in the Discussion section. Then, in the Materials and Methods section, we discuss, in detail, the mathematical modeling process.

**TABLE 1** Pertinent results of the variations in lanosterol concentration simulation

| Metabolites | Value in $\mu$M (%) | | | | | |
| --- | --- | --- | --- | --- | --- | --- |
| | Case 1 | Case 2 | Case 3 | Case 4 | Case 5 | Case 6 |
| Ergosterol | 0 (0%) | 4.61 (45.26%) | 45.48 (44.91%) | 65.41 (5.22%) | 40.38 (0.31%) | 9.69 (0.01%) |
| Toxic sterol | 0 (0%) | 3.03 (29.74%) | 29.89 (29.51%) | 222.01 (17.73%) | 644.09 (4.97%) | 1,066.38 (0.82%) |
| Percent ergosterol reduction | $-^a$ | 64.46% | 64.93% | 94.69% | 96.79% | 99.23% |
| Is ergosterol dominant? | – | Yes | Yes | No | No | No |
| Number of criteria satisfied | – | 2 | 2 | - | 1 | 1 |

*a*–, not applicable.

## RESULTS

The model was run to simulate several scenarios to investigate possible drug targets in the ergosterol pathway in *C. albicans*.

**Variations in lanosterol concentration.** The lanosterol concentration was varied as follows: 0×, 0.01×, 0.1×, 1×, 10×, and 100× the baseline value (see Fig. S2), represented by cases 1 to 6, respectively. Case 1 (0×) simulates a total inhibition of lanosterol. It results in the demise of *C. albicans*, as no ergosterol was produced, and ergosterol is essential for yeast survival. Cases 2 (0.01×) and 3 (0.1×) simulate the partial inhibition of lanosterol. They yield a higher percentage of the toxic sterol and ergosterol as the dominant metabolite but a lower percentage of ergosterol reduction, satisfying 2 out of 3 criteria which denote fair fluconazole resistance. Case 4 (1×) denotes the default case, which serves as our baseline. Cases 5 (10×) and 6 (100×) simulate the overproduction of lanosterol. They yield a lower percentage of the toxic sterol but a higher percentage of ergosterol reduction, satisfying 1 out of 3 criteria and denoting an equivocal result. Table 1 summarizes the results pertinent to the criteria for fluconazole resistance.

**Variations in C14$\alpha$-demethylase concentration (Erg11p).** The C14$\alpha$-demethylase enzyme concentration was varied as follows: 0×, 0.01×, 0.1×, 1×, 10×, and 100× the baseline value (see Fig. S3), represented by cases 1 to 6, respectively. Case 1 (0×) simulates a total inhibition of C14$\alpha$-demethylase. This results in the demise of *C. albicans*, as no ergosterol was produced, and ergosterol is essential for yeast survival. Cases 2 (0.01×) and 3 (0.1×) simulate the partial inhibition of C14$\alpha$-demethylase. They yield a higher percentage of toxic sterol and a higher percentage of ergosterol reduction, satisfying 0 out of 3 criteria, which denotes a good target for antifungals as an adjunct to fluconazole. Case 4 (1×) denotes the default case, which serves as our baseline. Cases 5 (10×) and 6 (100×) simulate the overexpression of the *ERG11* gene. They yield a lower percentage of the toxic sterol and a lower percentage of ergosterol reduction, satisfying 2 out of 3 criteria and denoting fair fluconazole resistance. Table 2 summarizes the results pertinent to the criteria for fluconazole resistance.

**Variations in fluconazole concentration.** The fluconazole concentration was varied as follows: 0×, 0.01×, 0.1×, 1×, 10×, and 100× the baseline value (see Fig. S4), represented by cases 1 to 6, respectively. Case 1 (0×) simulates a situation where no fluconazole was given. This results in the proliferation of *C. albicans*, as the dominant metabolite is ergosterol, and ergosterol is essential for yeast survival. Cases 2 (0.01×) and 3 (0.1×) simulate giving fluconazole with a dose lower than its minimum inhibitory concentration (MIC) (i.e., 2 $\mu$g/mL). They yield a lower percentage of toxic sterol and a

**TABLE 2** Pertinent results of the variations in C14$\alpha$-demethylase concentration simulation

| Metabolites | Value in $\mu$M (%) | | | | | |
| --- | --- | --- | --- | --- | --- | --- |
| | Case 1 | Case 2 | Case 3 | Case 4 | Case 5 | Case 6 |
| Ergosterol | 0 (0%) | 0.11 (0.01%) | 3.43 (0.27%) | 65.41 (5.22%) | 79.60 (6.36%) | 81.95 (6.55%) |
| Toxic sterol | 589.71 (46.17%) | 589.63 (46.17%) | 587.18 (46.08%) | 222.01 (17.73%) | 31.16 (2.49%) | 3.64 (0.29%) |
| Percent ergosterol reduction | $-^a$ | 99.68% | 98.91% | 94.69% | 93.74% | 93.59% |
| Is ergosterol dominant? | – | No | No | No | No | No |
| Number of criteria satisfied | – | 0 | 0 | – | 2 | 2 |

*a*–, not applicable.

**TABLE 3** Pertinent results of the variations in fluconazole concentration simulation

| Metabolites | Value in $\mu$M (%) | | | | | |
| --- | --- | --- | --- | --- | --- | --- |
| | Case 1 | Case 2 | Case 3 | Case 4 | Case 5 | Case 6 |
| Ergosterol | 1,230.86 (98%) | 66.73 (5.33%) | 66.62 (5.32%) | 65.41 (5.22%) | 3.89 (0.31%) | 0.11 (0.01%) |
| Toxic sterol | 0 (0%) | 204.17 (16.31%) | 205.67 (16.43%) | 222.01 (17.73%) | 586.84 (46.06%) | 589.62 (46.17%) |
| Percent ergosterol reduction | $-^a$ | 94.58% | 94.59% | 94.69% | 99.68% | 99.99% |
| Is ergosterol dominant? | – | No | No | No | No | No |
| Number of criteria satisfied | – | 2 | 2 | – | 0 | 0 |

$^a$–, not applicable.

lower percentage of ergosterol reduction, satisfying 2 out of 3 criteria, which denotes fair fluconazole resistance. Case 4 (1×) denotes the default case, which serves as our baseline. Cases 5 (10×) and 6 (100×) simulate giving fluconazole with a dose higher than its MIC. They yield a higher percentage of toxic sterol and a higher percentage of ergosterol reduction, satisfying 0 out of 3 criteria and denoting a good target for antifungals as an adjunct to fluconazole. Table 3 summarizes the results pertinent to the criteria for fluconazole resistance.

**Variations in sterol-methyltransferase concentration (Erg6p).** The sterol-methyltransferase enzyme concentration was varied as follows: 0×, 0.01×, 0.1×, 1×, 10×, and 100× the baseline value (see Fig. S5), represented by cases 1 to 6, respectively. Case 1 (0×) simulates a total inhibition of sterol-methyltransferase. This results in the demise of *C. albicans*, as no ergosterol was produced, and ergosterol is essential for yeast survival. Cases 2 (0.01×) and 3 (0.1×) simulate the partial inhibition of sterol-methyltransferase. They yield a lower percentage of toxic sterol and a lower percentage of ergosterol reduction, satisfying 2 out of 3 criteria, which denotes fair fluconazole resistance. Case 4 (1×) denotes the default case, which serves as our baseline. Cases 5 (10×) and 6 (100×) simulate the overexpression of the *ERG6* gene. They yield a higher percentage of toxic sterol and a higher percentage of ergosterol reduction, satisfying 0 out of 3 criteria and denoting a good target for antifungals as an adjunct to fluconazole. Table 4 summarizes the results pertinent to the criteria for fluconazole resistance.

**Variations in C5-desaturase concentration (Erg3p).** The C5-desaturase enzyme concentration was varied as follows: 0×, 0.01×, 0.1×, 1×, 10×, and 100× the baseline value (see Fig. S6), represented by cases 1 to 6, respectively. Case 1 (0×) simulates a total inhibition of C5-desaturase. This results in the demise of *C. albicans*, as no ergosterol was produced, and ergosterol is essential for yeast survival. Cases 2 (0.01×) and 3 (0.1×) simulate the partial inhibition of C5-desaturase. They yield a lower percentage of the toxic sterol but a higher percentage of ergosterol reduction, satisfying 1 out of 3 criteria, which denotes an equivocal result. Case 4 (1×) denotes the default case, which serves as our baseline. Cases 5 (10×) and 6 (100×) simulate the overexpression of the *ERG3* gene. They yield a higher percentage of the toxic sterol but a lower percentage of ergosterol reduction and ergosterol as the dominant metabolite, satisfying 2 out of 3 criteria and denoting fair fluconazole resistance. Table 5 summarizes the results pertinent to the criteria for fluconazole resistance.

**Sensitivity analysis.** The local one-at-a-time sensitivity analysis was plotted using a tornado chart for ergosterol, as shown in Fig. 2, and for toxic sterol, as shown in Fig. 3. The longer the bar a parameter has, the more sensitive the products (ergosterol and toxic sterol) are to it.

**TABLE 4** Pertinent results of the variations in sterol-methyltransferase concentration simulation

| Metabolites | Value in $\mu$M (%) | | | | | |
| --- | --- | --- | --- | --- | --- | --- |
| | Case 1 | Case 2 | Case 3 | Case 4 | Case 5 | Case 6 |
| Ergosterol | 0 (0%) | 9.31 (0.72%) | 58.54 (4.67%) | 65.41 (5.22%) | 46.52 (3.71%) | 15.07 (1.18%) |
| Toxic sterol | 0 (0%) | 3.63 (0.28%) | 34.58 (2.76%) | 222.01 (17.73%) | 660.83 (52.70%) | 796.76 (62.32%) |
| Percent ergosterol reduction | $-^a$ | 77.83% | 85.13% | 94.69% | 96.32% | 98.81% |
| Is ergosterol dominant? | – | No | No | No | No | No |
| Number of criteria satisfied | – | 2 | 2 | – | 0 | 0 |

$^a$–, not applicable.

**TABLE 5** Pertinent results of the variations in C5-desaturase concentration simulation

| Metabolites | Value in $\mu$M (%) | | | | | |
|---|---|---|---|---|---|---|
| | Case 1 | Case 2 | Case 3 | Case 4 | Case 5 | Case 6 |
| Ergosterol | 0 (0%) | 0.51 (0.04%) | 5.37 (0.41%) | 65.41 (5.22%) | 431.35 (44.26%) | 432.11 (44.34%) |
| Toxic sterol | 0 (0%) | 5.87 (0.45%) | 52.20 (4.02%) | 222.01 (17.73%) | 266.99 (27.39%) | 267.00 (27.40%) |
| Percent ergosterol reduction | $-^{a}$ | 96.03% | 95.81% | 94.69% | 66.36% | 66.31% |
| Is ergosterol dominant? | – | No | No | No | Yes | Yes |
| Number of criteria satisfied | – | 1 | 1 | – | 2 | 2 |

$^{a}$–, not applicable.

## DISCUSSION

The simulation for variations in lanosterol concentration showed that its total inhibition, shown in Table 1, case 1, by *ERG1* gene deletion results in the demise of *C. albicans* due to its lack of ergosterol. This agrees with the results of the study by Pasrija et al. (2005) that the *erg1* mutant showed a total lack of ergosterol and led to increased azole susceptibility (8). However, its partial inhibition, shown in Table 1, cases 2 and 3, results in fair fluconazole resistance. An example of this is the antifungal allylamine, which inhibits the squalene epoxidase enzyme in the pathway upstream of lanosterol. However, Kane and Carter showed that there is a synergistic effect when combining terbinafine (under allylamines) and fluconazole against *C. albicans* (9). Allylamines cause fungal cell death by depleting the ergosterol content and by accumulating squalene (10). The discrepancy may be due to our model being limited to the three implicated genes: *ERG11*, *ERG3*, and *ERG6*. Moreover, the overproduction of lanosterol, as shown in Table 1, cases 5 and 6, yields an equivocal result. Hence, no conclusion can be drawn. This indicates the need for further experiments or for further extension of the model to include upstream pathways, such as the squalene epoxidase enzyme.

The simulation for variations in C14$\alpha$-demethylase concentration confirmed that its partial inhibition, as shown in Table 2, cases 2 and 3, is a good target of antifungals, as it should be, since it has been known that this enzyme is the target of azole drugs, such as fluconazole, leading to the demise of *C. albicans* (11, 12). On the other hand, the overexpression of the *ERG11* gene is a known azole resistance mechanism (13). This has been confirmed by our simulation, shown in Table 2, cases 5 and 6, in that there was a lower percentage of toxic sterol and a lower percentage of ergosterol reduction.

The simulation for variations in fluconazole concentration, as shown in Table 3, confirmed that the effect of fluconazole in *C. albicans* is dose-dependent such that doses lower than its MIC do not fully eradicate the fungi, whereas doses greater than its MIC

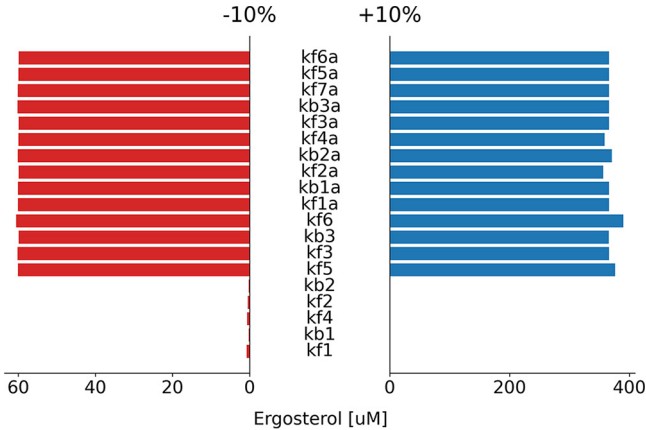

**FIG 2** Tornado chart of the local one-at-a-time sensitivity analysis of the model parameters to ergosterol. The *y* axis enumerates all of the model parameters, and the *x* axis represents the absolute difference of the resulting ergosterol concentration to its baseline value. The red bars indicate the sensitivity analysis of the −10% of the baseline values, whereas the blue bars indicate the sensitivity analysis of the +10% of the baseline values. The longer the bars for the parameters are, the more sensitive the ergosterol is to the parameters. See Table S1 for the parameters used in the study.

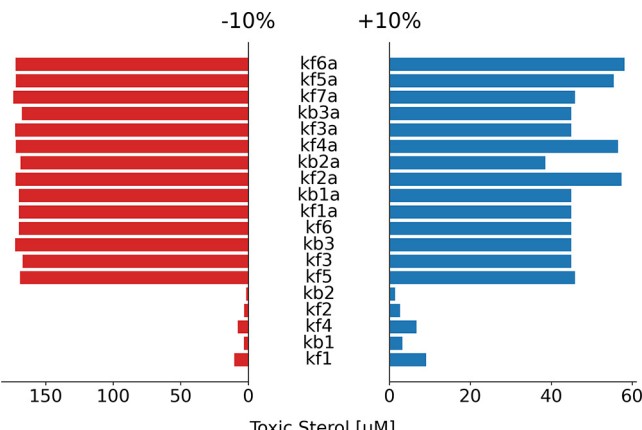

**FIG 3** Tornado chart of the local one-at-a-time sensitivity analysis of the model parameters to the toxic sterol. The *y* axis enumerates all of the model parameters, and the *x* axis represents the absolute difference of the resulting toxic sterol concentration to its baseline value. The red bars indicate the sensitivity analysis of the −10% of the baseline values, whereas the blue bars indicate the sensitivity analysis of the +10% of the baseline values. The longer the bars for the parameters are, the more sensitive the ergosterol is to the parameters. See Table S1 for the parameters used in the study.

do eradicate them. Hence, it is crucial to have an accurate quantitative method by which to determine the MIC for antifungal susceptibility testing (14).

The simulation for variations in sterol-methyltransferase concentration showed that its partial inhibition, as shown in Table 4, cases 2 and 3, yields fair fluconazole resistance. This agrees with the findings of Kelly et al. (1997) that show that a defect in the *ERG6* gene leads to toxic sterol accumulation and, hence, the resistance to fluconazole (15). This resistance mechanism is also described in the review by Bhattacharya et al. (2020) (12). On the other hand, our results showed that the overexpression of the *ERG6* gene, shown in Table 4, cases 5 and 6, is a good target for antifungals as an adjunct to fluconazole. Bhattacharya et al. (2018) noted that in *Saccharomyces cerevisiae*, overexpression of the *ERG6* gene encourages the initiation of the alternative pathway, leading to the production of toxic sterol (16). In the review by Bhattacharya et al. (2020), the *ERG6* gene contributes to the formation of the toxic sterol in the presence of azoles (12). Hence, we can infer that the overexpression of the *ERG6* gene can lead to an increase in the formation of the toxic sterol and can result in a hypersusceptibility to azoles. As such, this can have an additive effect when combined with fluconazole to circumvent the resistance of *C. albicans* against it. The overexpression of the *ERG6* gene yields a higher percentage of toxic sterol and a higher percentage of ergosterol reduction. Hence, we recommend conducting further experiments on the sterol-methyltransferase enzyme to confirm that its overexpression is a possible drug target as an adjunct to fluconazole in order to validate our simulation result.

The simulation for variations in C5-desaturase concentration showed that its partial inhibition, as shown in Table 5, cases 2 and 3, leads to an equivocal result. Hence, no conclusion can be drawn. However, in the literature, it is known that its mutation is linked with azole resistance (5, 12, 17). On the other hand, our simulation of the overexpression of the *ERG3* gene, shown in Table 5, cases 5 and 6, results in fair fluconazole resistance. No studies have yet been conducted regarding this in the literature; however, our results showed that it yields a lower percentage of ergosterol reduction and ergosterol as the dominant metabolite. Hence, we recommend against looking into this enzyme as a possible drug target as an adjunct to fluconazole.

The local one-at-a-time sensitivity analysis, as shown in Fig. 2 and 3, showed that the more upstream the pathway is, the less sensitive the products (ergosterol and toxic sterol) are to them, whereas the more downstream the pathway is, the more sensitive the products are to them. Also, the products (ergosterol and toxic sterol) are more sensitive to the parameters in the alternative pathway than to those in the normal pathway.

**Concluding remarks.** From the model simulations, we found the following results: (i) a partial inhibition of sterol-methyltransferase yields a fair amount of fluconazole resistance; (ii) overexpression of the *ERG6* gene, leading to an increased sterol-methyltransferase enzyme, is a good target of antifungals as an adjunct to fluconazole; (iii) a partial inhibition of lanosterol yields a fair amount of fluconazole resistance; (iv) C5-desaturase is not a good target of antifungals as an adjunct to fluconazole; (v) C14$\alpha$-demethylase is confirmed to be a good target for fluconazole; and (vi) the dose-dependent effect of fluconazole is confirmed. This study hopes to aid experimenters in narrowing down the possible drug targets prior to costly and time-consuming experiments and to serve as a cross-validation tool for experimental data.

We recommend extending the model to include other genes involved in the ergosterol biosynthesis pathway, such as the *ERG1* gene that expresses squalene epoxidase, which is the target of antifungal allylamine. Moreover, the computational approach used in our study is also applicable to other, completely different pathways involved in the azole resistance mechanism, such as the *CDR1*, *CDR2*, and *MDR1* genes, which are involved in the drug efflux transporter. We also recommend doing further experiments on the sterol-methyltransferase enzyme to confirm that its overexpression is a possible target for antifungals as an adjunct to fluconazole in order to validate our simulation result.

## MATERIALS AND METHODS

The model for the ergosterol biosynthesis pathway, as shown in Fig. 4, in *C. albicans* is built by adopting the pathways of Sanglard et al. (2003) (see Figure 1 in [5]), and Martel et al. (2010) (see Figure 1 in [17]). We noted several differences between the two studies, notably, the substrate for the C14$\alpha$-demethylase enzyme is lanosterol in Sanglard et al. (2003); however, it is eburicol in the pathway presented by Martel et al. (2010). There are still disparities in the literature regarding this matter. In this study, we followed Sanglard et al. (2003) for the construction of our model. The Python programming language was used for the coding of the model equations and simulations (18).

Due to the complexity of the calculations, the model is limited to the three genes implicated in azole resistance in the literature: *ERG11*, *ERG3*, and *ERG6*. Various point mutations in the *ERG11* gene were shown to alter the binding abilities of azoles to the C14$\alpha$-demethylase enzyme (19, 20). The deletion of the *ERG3* gene resulted in the accumulation of the toxic sterol precursor 14$\alpha$-methylfecosterol instead of the toxic sterol, allowing for *C. albicans* survival (15, 17). Disruption in the *ERG6* gene was shown to cause hypersusceptible strains to antifungal agents but not to azoles (21). Moreover, Akins (2005) noted that the *ERG6* gene is an attractive target for antifungals (22). Hence, we deem it worthwhile to investigate whether or not it has an additive effect with fluconazole.

Another limitation of the study is that our simulations are limited to testing drug targets as an adjunct to fluconazole, since the model calibration was based on the study by Kelly et al. (1997), which compared the sterol compositions of *C. albicans* before and after fluconazole treatment using gas chromatography/mass spectrometry (GC/MS) (15).

The model uses ordinary differential equations (ODEs) with mass-action kinetics, as first described in (23), which has since become a standard method for mathematical modeling. Recently, Sutradhar et al. (2021) used an ODE model to determine the drivers of the growth of antibiotic-resistant bacteria in wastewater (24). The ODEs for the model without azole (normal pathway) are shown in Equation 1, as:

$$\frac{dR_1}{dt} = k_{-1}C_1 - k_{-2a}C_{2a} - k_1R_1E_1 - k_{2a}R_1E_2,$$

$$\frac{dE_1}{dt} = k_{-1}C_1 + k_4C_1 + k_{-1a}C_{1a} - k_1R_1E_1 - k_{1a}R_{1a}E_1,$$

$$\frac{dC_1}{dt} = k_1R_1E_1 - k_{-1}C_1 - k_4C_1,$$

$$\frac{dR_2}{dt} = k_4C_1 + k_{-2}C_2 - k_2R_2E_2,$$

$$\frac{dE_2}{dt} = k_{-2}C_2 + k_5C_2 + k_{-2a}C_{2a} + k_{4a}C_{2a} - k_2R_2E_2 - k_{2a}R_1E_2,$$

$$\frac{dC_2}{dt} = k_2R_2E_2 - k_{-2}C_2 - k_5C_2,$$

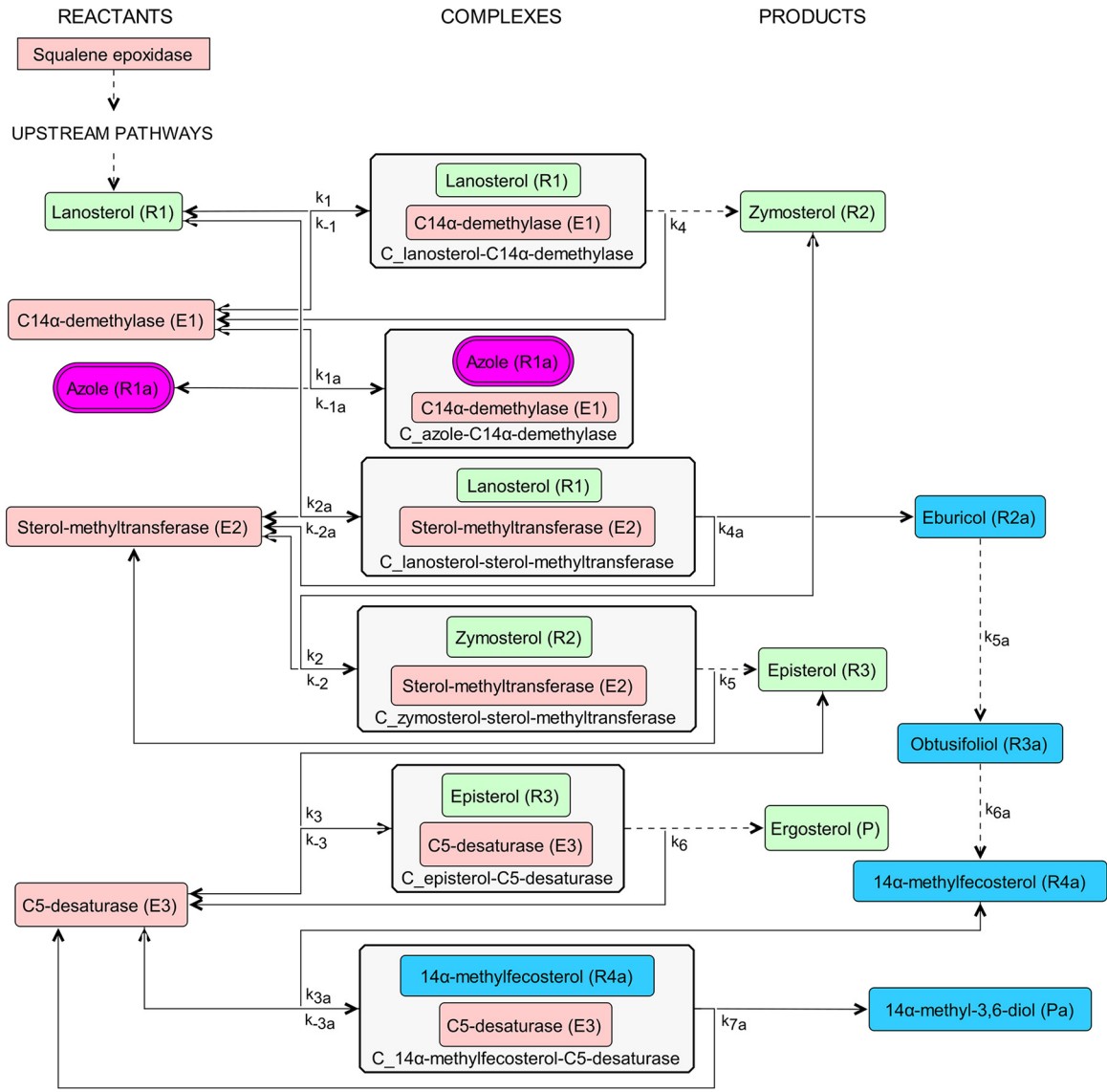

**FIG 4** Ergosterol biosynthesis pathway. The pathway starts from lanosterol to ergosterol synthesis. It does not include the pathways upstream of lanosterol, such as the squalene epoxidase enzyme. The metabolites in green denote the ergosterol pathway in *C. albicans* when azole is absent, which leads to its survival, whereas those in blue represent the alternative pathway during azole inhibition, which leads to its demise, due to the production of the toxic sterol, $14\alpha$-methyl-3,6-diol. The enzymes are colored in pink. The azole drug is colored in magenta. The arrows pertain to the direction of the reaction. Bidirectional arrows indicate reversible reactions.

$$\frac{dR_3}{dt} = k_5 C_2 + k_{-3} C_3 - k_3 R_3 E_3,$$

$$\frac{dE_3}{dt} = k_{-3} C_3 + k_6 C_3 + k_{-3a} C_{3a} + k_{7a} C_{3a} - k_3 R_3 E_3 - k_{3a} R_{4a} E_3,$$

$$\frac{dC_3}{dt} = k_3 R_3 E_3 - k_{-3} C_3 - k_6 C_3,$$

$$\frac{dP}{dt} = k_6 C_3. \tag{1}$$

The ODEs for the model with azole (alternative pathway) are shown in Equation 2, as:

$$\frac{dR_{1a}}{dt} = k_{-1a} C_{1a} - k_{1a} R_{1a} E_1,$$

$$\frac{dC_{1a}}{dt} = k_{1a}R_{1a}E_1 - k_{-1a}C_{1a},$$

$$\frac{dC_{2a}}{dt} = k_{2a}R_1E_2 - k_{-2a}C_{2a} - k_{4a}C_{2a},$$

$$\frac{dR_{2a}}{dt} = k_{4a}C_{2a} - k_{5a}R_{2a},$$

$$\frac{dR_{3a}}{dt} = k_{5a}R_{2a} - k_{6a}R_{3a},$$

$$\frac{dR_{4a}}{dt} = k_{6a}R_{3a} + k_{-3a}C_{3a} - k_{3a}R_{4a}E_3,$$

$$\frac{dC_{3a}}{dt} = k_{3a}R_{4a}E_3 - k_{-3a}C_{3a} - k_{7a}C_{3a},$$

$$\frac{dP_a}{dt} = k_{7a}C_{3a}. \tag{2}$$

The constraint equations are shown in Equation 3, as:

$$E_1 = E_{1,total} - C_1 - C_{1a},$$

$$E_2 = E_{2,total} - C_2 - C_{2a},$$

$$E_3 = E_{3,total} - C_3 - C_{3a}. \tag{3}$$

In the study by Hargrove et al. (2017), they used a molar ratio of 1:2:50 for the enzyme, inhibitor, and substrate, respectively (25). We tested it in our model and observed that this enzyme concentration was insufficient for the inhibitor to fully take its effect. As such, we adjusted the assignment of the molar ratio to 4:2:50 for the enzyme, inhibitor, and substrate, respectively. The fluconazole concentration is set at 16 $\mu$g/mL, following Kelly et al. (1997) (15). The parameter values, shown in Table S1, were obtained by calibrating the model to the sterol compositions described by Kelly et al. (1997) (15) such that in a fluconazole-sensitive *C. albicans* strain, it is mainly composed of ergosterol (98%), while the rest is classified as unknown (2%). We distribute the unknown 2% to the remaining metabolites in the pathway. In a fluconazole-resistant strain, the sterol composition is as follows: ergosterol (2%), eburicol (16.1%), obtusifoliol (34.5%), 14$\alpha$-methyl-3,6-diol (45.2%), and unknown (2.2%). We again distribute the unknown 2.2% to the remaining metabolites in the pathway (see Table 8 in [15]). Following Kelly et al. (1997), the simulations were run for 24 h, and the pertinent results were tabulated thereafter. The model generated the plots shown in Fig. S1. The pertinent results are shown in Table 6.

The clinical breakpoints, as determined from the 24-h Clinical & Laboratory Standards Institute (CLSI) M27-A3 (26) broth microdilution method, for fluconazole against *C. albicans* are as follows: Susceptible (MIC $\leq$ 2 $\mu$g/mL), Dose-dependent (MIC = 4 $\mu$g/mL), and Resistant (MIC $\geq$ 8 $\mu$g/mL) (14). We recalibrated our model to accommodate the MIC breakpoint for fluconazole susceptibility (MIC = 2 $\mu$g/mL) to determine its threshold for fluconazole susceptibility. To compute the molar concentration of 2 $\mu$g/mL, we used the following calculation (molecular weight of fluconazole: 306.3 g/mol from [6]):

$$\frac{2\,\mu g}{1\,mL} \times \frac{1\,mol}{306.3\,g} \times \frac{1000\,mL}{1\,L} = 6.53\,\mu M. \tag{4}$$

The recalibrated model generated the plots shown in Fig. 5. This served as the baseline for the simulations in the Results section. The pertinent results are shown in Table 7.

A local one-at-a-time sensitivity analysis was performed on the model parameters. We took $-10\%$ and $+10\%$ of the baseline values of the parameters, as shown in Table S1. Then, we gathered the absolute differences of the product (ergosterol and toxic sterol) concentrations from their baseline values.

**TABLE 6** Pertinent results of the model calibration to that of Kelly et al. (1997) (15)

| | Value in $\mu$M (%) | |
|---|---|---|
| Metabolites | (−) Azole | (+) Azole |
| Ergosterol | 1,230.86 (98%) | 25.14 (2%) |
| 14$\alpha$-methyl-3,6-diol | 0 (0%) | 567.29 (45.20%) |
| Percent ergosterol reduction | −$^a$ | 97.96% |

$^a$−, not applicable.

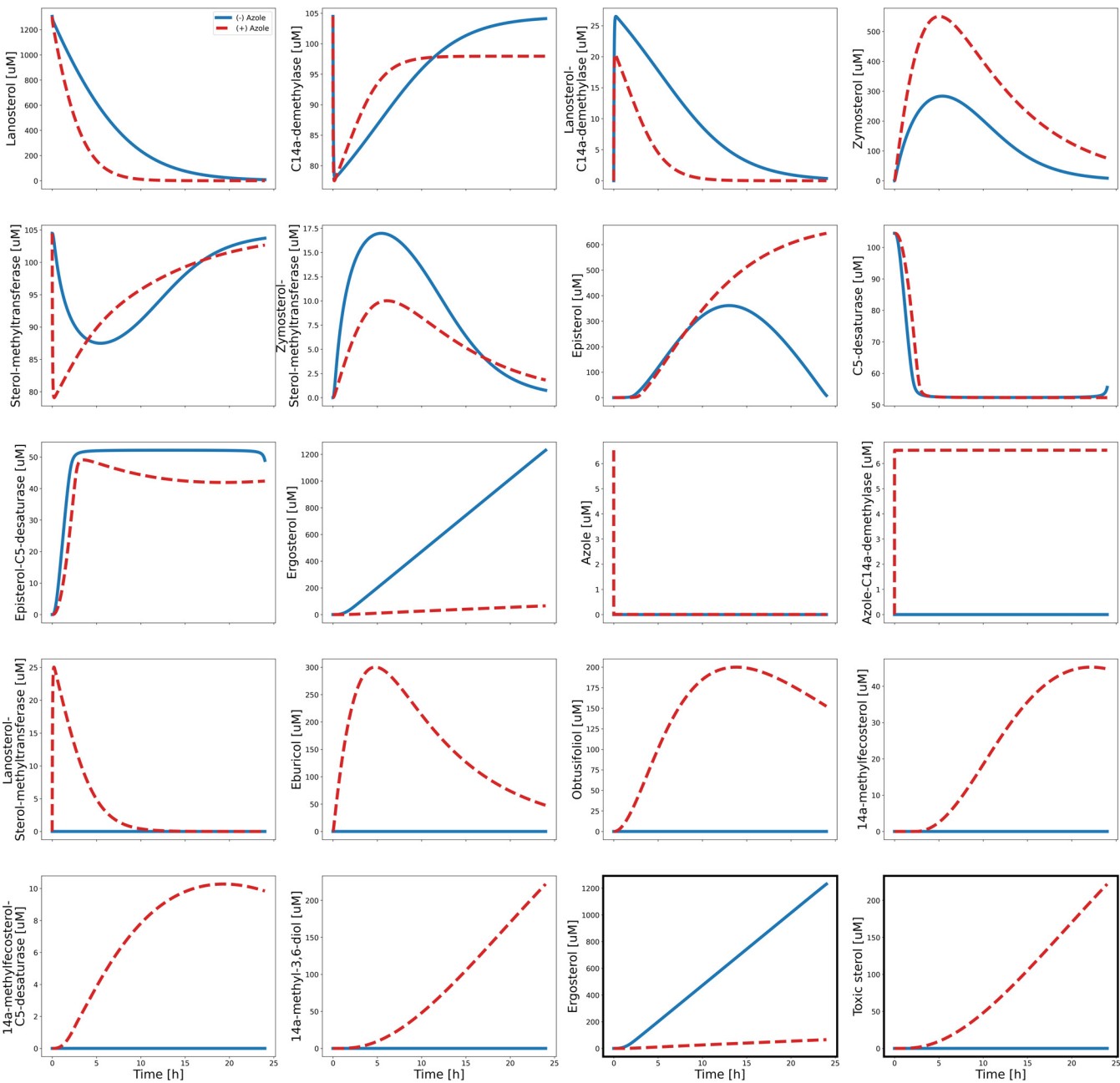

**FIG 5** Plots generated by the model recalibrated to the MIC breakpoint for fluconazole susceptibility. The normal pathway, when azole is absent, is denoted by blue solid lines. The alternative pathway, when azole is present, is denoted by red dashed lines. The *x* axes denote the time in hours, and the *y* axes represent the concentrations in $\mu$M. The last two subplots, emphasized in dark borders, reiterate the pertinent plots of ergosterol and the toxic sterol (14$\alpha$-methyl-3,6-diol) for easier comparison.

In experimental studies, antifungal susceptibility is quantified via a visual spotting assay on yeast extract peptone dextrose (YEPD) medium (5), the disc diffusion method (19), or the broth microdilution method (27). However, these methods are not feasible to use in mathematical models because they cannot be quantified. As such, we looked for quantitative methods at the metabolite level to circumvent this limitation.

Arthington-Skaggs et al. (1999) proposed the use of percent ergosterol reduction as a quantitative method by which to evaluate antifungal susceptibility (28). However, Kelly et al. (1997) found that an accumulation of the toxic sterol causes the growth arrest of *C. albicans* (15). Another interesting result found by Martel et al. (2010) is that antifungal susceptibility holds when the dominant fraction of sterol is ergosterol, regardless of the amount of the toxic sterol (17).

Considering the above methods for antifungal susceptibility, we propose the following criteria, based on the data gathered in Table 7, for the determination of the threshold for fluconazole resistance in *C. albicans*: (i) when the percent toxic sterol (14$\alpha$-methyl-3,6-diol) is less than 17.73%; (ii) when the percent ergosterol reduction is less than 94.69%; and (iii) when ergosterol is the dominant fraction among the metabolites.

**TABLE 7** Pertinent results of the model recalibration to the MIC breakpoint for fluconazole susceptibility

| Metabolites | Value in $\mu$M (%) | |
| --- | --- | --- |
| | (−) azole | (+) azole |
| Ergosterol | 1,230.95 (98%) | 65.41 (5.22%) |
| 14$\alpha$-methyl-3,6-diol | 0 (0%) | 222.01 (17.73%) |
| Percent ergosterol reduction | −[a] | 94.69% |

[a]−, not applicable.

When all three criteria are satisfied, we designate it as good fluconazole resistance. When two out of three are satisfied, we designate it as fair fluconazole resistance. When only one out of three is satisfied, we designate it as an equivocal result such that no conclusion can be drawn. Conversely, when none of the criteria are satisfied, we designate it as a good target for antifungals as an adjunct to fluconazole.

**Data availability.** The source code is available at https://github.com/dlsu-scomb/candida-amr.

## SUPPLEMENTAL MATERIAL

Supplemental material is available online only.

**FIG S1**, PDF file, 0.04 MB.
**FIG S2**, PDF file, 0.1 MB.
**FIG S3**, PDF file, 0.1 MB.
**FIG S4**, PDF file, 0.1 MB.
**FIG S5**, PDF file, 0.1 MB.
**FIG S6**, PDF file, 0.05 MB.
**TABLE S1**, PDF file, 0.05 MB.

## ACKNOWLEDGMENTS

P.K.Y. would like to thank the Department of Science and Technology - Science Education Institute (DOST-SEI) and Leave a Nest Philippines, Inc. for their support. The research was conducted under the Leave a Nest Research Grant. We also thank the Department of Science and Technology - Philippine Council for Health Research and Development (DOST-PCHRD) and De La Salle University Science Foundation, Inc. for their support. The funding agencies did not have any additional role in the study design, data collection, analysis, decision to publish, or preparation of the manuscript.

We are grateful to the three anonymous referees whose comments helped greatly improve the manuscript.

All authors have read and agreed to the published version of the manuscript.

We declare no conflict of interest.

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
