## [Reviewer comments · mSystems]

Mathematical Modeling of Fluconazole Resistance in the Ergosterol Pathway of *Candida albicans*

Paul Yu, Llewelyn Moron-Espiritu, and Angelyn Lao

Corresponding Author(s): Angelyn Lao, De La Salle University

Review Timeline:

Submission Date:	July 22, 2022
Editorial Decision:	September 18, 2022
Revision Received:	September 30, 2022
Editorial Decision:	October 3, 2022
Revision Received:	October 4, 2022
Accepted:	October 23, 2022

Editor: Babak Momeni

Reviewer(s): Disclosure of reviewer identity is with reference to reviewer comments included in decision letter(s). The following individuals involved in review of your submission have agreed to reveal their identity: Begum Alaybeyoglu (Reviewer #1)

Transaction Report:

DOI: <https://doi.org/10.1128/msystems.00691-22>

September 18, 2022

Dr. Angelyn Relucio Lao
De La Salle University
Systems and Computational Biology Unit, Department of Mathematics and Statistics
2401 Taft Avenue, Malate
Manila, NCR 0922
Philippines

Re: mSystems00691-22 (Mathematical Modeling of Fluconazole Resistance in the Ergosterol Pathway of *Candida albicans*)

Dear Dr. Angelyn Relucio Lao:

Thank you for submitting your manuscript to mSystems. We have completed our review and I am pleased to inform you that, in principle, we expect to accept it for publication in mSystems. However, acceptance will not be final until you have adequately addressed the reviewer comments.

Preparing Revision Guidelines

Sincerely,

Babak Momeni

Editor, mSystems

Journals Department
American Society for Microbiology
1752 N St., NW

Reviewer comments:

Reviewer #1 (Comments for the Author):

This is a well written manuscript and would be of interest for the researchers in this field. Overall, the findings are interesting and will help the field move further in fighting with drug resistance.

I have 1 minor comment:

It is described on a high level how the parametrization of the model was performed using Kelly et al study. In that regard, authors should include a local/global sensitivity analysis to evaluate and finalize the model.

Reviewer #2 (Comments for the Author):

Azole resistance continues to be a challenge in treating Candidiasis. Yu et al. developed a mathematical model to predict and understand how resistance is built up by different genes in the pathway. To simplify calculations, ERG11, ERG3, and ERG6 of ergosterol pathway were selected. Sufficient experimental data have been accumulated in the field to understand the role of each gene, which streamlined the validation of the mathematical model. The authors were able to confirm previous experimental findings and offer new insights: while methyltransferase is a good target of antifungals, C5-desaturase is not. Using a computational model circumvents the need to carry out extensive wet lab work to confirm or refute hypothesis.

This work is creative and provides an alternative perspective to understand antifungal resistance. It would be great if the authors could discuss how applicable and generalizable this computational approach is toward other inhibitors on Candidiasis, and inhibition of completely different pathways. Furthermore, to cater to a broader audience, the authors are encouraged to include a biosynthetic pathway of ergosterol and the representative structures of azoles as a figure in the main text. This would make it easier for readers to follow the flow of the paper.

Reviewer #3 (Comments for the Author):

A. There are a lot of grammar mistakes:

- 1- Line 20 is reported to be not as.
- 2- Line 22 add the word 'The' before most common and delete it before candida in the same line.
- 3- Line 35 add 'an' before adjunct to fluconazole.
- 4- Line 38 replace narrow with narrowing down.
- 5- Line 43 add 'the' before the word most.
- 6- Line 44 remove 'the' before Candida albicans.
- 7- Line 61 remove 'the' before Candida albicans.
- 8- Line 67 add 'the' before the word invasive.
- 9- Line 73 remove the comma after the word epidemiology.
- 10- Line 84 transforms the word non-albicans into singular form 'non-albican'.
- 11- Line 91 replace in with into before the word azole.
- 12- Line 96 remove the comma after the word experiment.
- 13- Line 112 add 'the' before the word Python.
- 14- Line 118 add 'the' before the word Deletion.
- 15- Line 128 transforms the word "compares" from present tense to past tense.
- 16- Line 134 add 'the' before the word growth.
- 17- Line 200 transform the word case to plural form 'cases'.
- 18- Line 204 removes the comma after the word sterol.
- 19- Line 205 remove 'a' before the word fair.
- 20- Line 208 removes the comma after the word sterol.
- 21- Line 214 transform the word case to plural form 'cases'.
- 22- Line 216 removes the comma after the word albicans.
- 23- Line 218 removes the comma after the word sterol.
- 24- Line 220 add 'an' before the word adjunct.
- 25- Line 223 removes the comma after the word sterol.
- 26- Line 224 remove 'a' before the word fair.
- 27- Line 228 transform the word case to plural form 'cases'.

- 28- Line 230 removes the comma after the word albicans.
- 29- Line 231 remove 'a' before the word fluconazole.
- 30- Line 233 removes the comma after the word sterol.
- 31- Line 234 remove 'a' before the word fair.
- 32- Line 237 removes the comma after the word sterol.
- 33- Line 238 add 'an' before the word adjunct.
- 34- Line 243 transform the word case to plural form 'cases'.
- 35- Line 245 removes the comma after the word albicans.
- 36- Line 247 removes the comma after the word sterol.
- 37- Line 248 remove 'a' before the word fair.
- 38- Line 251 removes the comma after the word sterol.
- 39- Line 253 add 'an' before the word adjunct.
- 40- Line 257 transform the word case to plural form 'cases'.
- 41- Line 259 removes the comma after the word albicans.
- 42- Line 261 and 265 remove the comma after the word sterol.
- 43- Line 266 remove 'a' before the word fair.
- 44- Line 271 add comma after word deletion.
- 45- Line 275 remove 'a' before the word fair.
- 46- Line 279 remove comma after the word content and replace the word accumulation with accumulating.
- 47- Line 281 removes the word genes after ERG6.
- 48- Line 283 remove comma after the word experiment.
- 49- Line 290 remove 'the' after the word C. albicans and 'an' before the word overexpression.
- 50- Line 293 removes the comma after the word sterol.
- 51- Line 303 and 308 remove 'a' before the word fair.
- 52- Line 312 and 314 remove the comma after the word sterol.
- 53- Line 316 replace 'by' with 'of'.
- 54- Line 326 remove 'a' before the word fair.
- 55- Line 340 remove 'doing' before the word costly.
- 56- Line 346 add 'the' before the word azole.

B. Here are some missing details that need to be mentioned:

- 1- Mention more details about the methods used for the experimental antifungal assay.
- 2- Mention the CLSI version which was used as a reference in the antifungal assay.

POINT-BY-POINT RESPONSE

Reviewer comments:

Reviewer #1 (Comments for the Author):

This is a well written manuscript and would be of interest for the researchers in this field. Overall, the findings are interesting and will help the field move further in fighting with drug resistance.

I have 1 minor comment:

It is described on a high level how the parametrization of the model was performed using Kelly et al study. In that regard, authors should include a local/global sensitivity analysis to evaluate and finalize the model.

Sensitivity analysis was added in the following:

Lines 172-175:

172 A local one-at-a-time sensitivity analysis was performed on the model parameters. We
173 take the -10% and +10% of the baseline values of the parameters, as shown in Table
174 S1. We then gathered the absolute difference of the product (ergosterol and toxic sterol)
175 concentrations from their baseline values, respectively.

Lines 276-280:

276 *3.6 Sensitivity analysis*
277 The local one-at-a-time sensitivity analysis was plotted using a tornado chart for
278 ergosterol, as shown in Figure 4, and for toxic sterol, as shown in Figure 5. The longer
279 the bar a parameter has, the more sensitive the products (ergosterol and toxic sterol)
280 are to it.

Lines 343-347:

343 The local one-at-a-time sensitivity analysis, as shown in Figures 4 and 5, showed that
344 the more upstream the pathway is, the less sensitive the products (ergosterol and toxic
345 sterol) are to them. Whereas the more downstream the pathway is, the more sensitive
346 the products are to them. Also, the products (ergosterol and toxic sterol) are more
347 sensitive to the parameters in the alternative pathway than in the normal pathway.

Figure 4: (lines 542-548)

542 **Figure 4.** Tornado chart of the local one-at-a-time sensitivity analysis of the model
543 parameters to ergosterol. The y-axis enumerates all the model parameters while the x-
544 axis represents the absolute difference of the resulting ergosterol concentration to its
545 baseline value. The red bars indicate the sensitivity analysis of the -10% of the baseline
546 values. Whereas the blue bars indicate the sensitivity analysis of the +10% of the
547 baseline values. The longer the bars are the more sensitive the ergosterol is to them.
548 See Table S1 for the parameters used in the study.

Figure 5: (lines 551-557)

551 **Figure 5.** Tornado chart of the local one-at-a-time sensitivity analysis of the model
552 parameters to the toxic sterol. The y-axis enumerates all the model parameters while
553 the x-axis represents the absolute difference of the resulting toxic sterol concentration to
554 its baseline value. The red bars indicate the sensitivity analysis of the -10% of the
555 baseline values. Whereas the blue bars indicate the sensitivity analysis of the +10% of
556 the baseline values. The longer the bars are the more sensitive the ergosterol is to
557 them. See Table S1 for the parameters used in the study.

Reviewer #2 (Comments for the Author):

Azole resistance continues to be a challenge in treating Candidiasis. Yu et al. developed a mathematical model to predict and understand how resistance is built up by different genes in the pathway. To simplify calculations, ERG11, ERG3, and ERG6 of ergosterol pathway were selected. Sufficient experimental data have been accumulated in the field to understand the role of each gene, which streamlined the validation of the mathematical model. The authors were able to confirm previous experimental findings and offer new insights: while methyltransferase is a good target of antifungals, C5-desaturase is not. Using a computational model circumvents the need to carry out extensive wet lab work to confirm or refute hypothesis.

This work is creative and provides an alternative perspective to understand antifungal resistance. It would be great if the authors could discuss how applicable and generalizable this computational approach is toward other inhibitors on Candidiasis, and inhibition of completely different pathways.

The generalizability of the approach is added to lines 362-363.

362 which is the target of allylamine antifungals. Moreover, the computational approach
363 used in our study is also applicable to other completely different pathways involved in
364 the azole resistance mechanism, such as the *CDR1*, *CDR2*, and *MDR1* genes, which
365 are involved in the drug efflux transporter. We also recommend doing further

Furthermore, to cater to a broader audience, the authors are encouraged to include a biosynthetic pathway of ergosterol and the representative structures of azoles as a figure in the main text. This would make it easier for readers to follow the flow of the paper.

The ergosterol biosynthesis pathway is shown in Figure 2.

Whereas the fluconazole structure is now added as Figure 1.

Reviewer #3 (Comments for the Author):

A. There are a lot of grammar mistakes:

- 1- Line 20 is reported to be not as.
- 2- Line 22 add the word 'The' before most common and delete it before candida in the same line.
- 3- Line 35 add 'an' before adjunct to fluconazole.
- 4- Line 38 replace narrow with narrowing down.
- 5- Line 43 add 'the' before the word most.
- 6- Line 44 remove 'the' before Candida albicans.
- 7- Line 61 remove 'the' before Candida albicans.
- 8- Line 67 add 'the' before the word invasive.
- 9- Line 73 remove the comma after the word epidemiology.
- 10- Line 84 transforms the word non-albicans into singular form 'non-Albican'.
- 11- Line 91 replace in with into before the word azole.
- 12- Line 96 remove the comma after the word experiment.
- 13- Line 112 add 'the' before the word Python.
- 14- Line 118 add 'the' before the word Deletion.
- 15- Line 128 transforms the word "compares" from present tense to past tense.
- 16- Line 134 add 'the' before the word growth.
- 17- Line 200 transform the word case to plural form 'cases'.
- 18- Line 204 removes the comma after the word sterol.
- 19- Line 205 remove 'a' before the word fair.
- 20- Line 208 removes the comma after the word sterol.
- 21- Line 214 transform the word case to plural form 'cases'.
- 22- Line 216 removes the comma after the word albicans.
- 23- Line 218 removes the comma after the word sterol.
- 24- Line 220 add 'an' before the word adjunct.
- 25- Line 223 removes the comma after the word sterol.
- 26- Line 224 remove 'a' before the word fair.
- 27- Line 228 transform the word case to plural form 'cases'.
- 28- Line 230 removes the comma after the word albicans.
- 29- Line 231 remove 'a' before the word fluconazole.
- 30- Line 233 removes the comma after the word sterol.
- 31- Line 234 remove 'a' before the word fair.
- 32- Line 237 removes the comma after the word sterol.
- 33- Line 238 add 'an' before the word adjunct.
- 34- Line 243 transform the word case to plural form 'cases'.
- 35- Line 245 removes the comma after the word albicans.
- 36- Line 247 removes the comma after the word sterol.
- 37- Line 248 remove 'a' before the word fair.

- 38- Line 251 removes the comma after the word sterol.
- 39- Line 253 add 'an' before the word adjunct.
- 40- Line 257 transform the word case to plural form 'cases'.
- 41- Line 259 removes the comma after the word albicans.
- 42- Line 261 and 265 remove the comma after the word sterol.
- 43- Line 266 remove 'a' before the word fair.
- 44- Line 271 add comma after word deletion.
- 45- Line 275 remove 'a' before the word fair.
- 46- Line 279 remove comma after the word content and replace the word accumulation with accumulating.
- 47- Line 281 removes the word genes after ERG6.
- 48- Line 283 remove comma after the word experiment.
- 49- Line 290 remove 'the' after the word C. albicans and 'an' before the word overexpression.
- 50- Line 293 removes the comma after the word sterol.
- 51- Line 303 and 308 remove 'a' before the word fair.
- 52- Line 312 and 314 remove the comma after the word sterol.
- 53- Line 316 replace 'by' with 'of'.
- 54- Line 326 remove 'a' before the word fair.
- 55- Line 340 remove 'doing' before the word costly.
- 56- Line 346 add 'the' before the word azole.

We have accordingly changed all the grammatical corrections described above.

B. Here are some missing details that need to be mentioned:

1- Mention more details about the methods used for the experimental antifungal assay.

We have indicated in the methods section the different assays for antifungal susceptibility on lines 177 - 181.

177 In experimental studies, antifungal susceptibility is quantified by visual spotting assay on
178 YEPD medium, such as in [5], disc diffusion method, such as in [10], or broth
179 microdilution method, such as in [20]. However, these methods are not feasible to use
180 in mathematical models since they cannot be quantified. As such, we look for
181 quantitative methods at the metabolite level to circumvent this limitation.

The assays were not performed in the study since the focus of our research is on the modeling approach.

2- Mention the CLSI version which was used as a reference in the antifungal Assay.

Please see Line 161 for the changes.

160 The clinical breakpoints as determined from the 24-hour Clinical & Laboratory
161 Standards Institute (CLSI) M27-A3 [18] broth microdilution method for fluconazole
162 against *C. albicans* are as follows: Susceptible (minimum inhibitory concentration (MIC)
163 $\leq 2 \mu\text{g/ml}$), Dose-dependent (MIC = $4 \mu\text{g/ml}$), and Resistant (MIC $\geq 8 \mu\text{g/ml}$) [19]. We re-

October 3, 2022

Dr. Angelyn Relucio Lao
De La Salle University
Department of Mathematics and Statistics
2401 Taft Avenue, Malate
Manila, NCR 0922
Philippines

Re: mSystems00691-22R1 (Mathematical Modeling of Fluconazole Resistance in the Ergosterol Pathway of *Candida albicans*)

Dear Dr. Angelyn Relucio Lao:

Thank you for submitting your manuscript to mSystems. I am pleased to inform you that, in principle, we expect to accept it for publication in mSystems. However, acceptance will not be final until you have adequately addressed the following minor comments.

1. In Figures 3, S1-S6, please increase the font sizes for the labels of axes to make them more visible.
2. In Figures 4 and 5, please include a label and units for the x-axis.
3. Please include a statement regarding the availability of codes and data generated and used in the manuscript.

Preparing Revision Guidelines

Sincerely,

Babak Momeni

Editor, mSystems

Journals Department
Reviewer comments:

POINT-BY-POINT RESPONSE

Reviewer comments:

Reviewer #1 (Comments for the Author):

This is a well written manuscript and would be of interest for the researchers in this field. Overall, the findings are interesting and will help the field move further in fighting with drug resistance.

I have 1 minor comment:

It is described on a high level how the parametrization of the model was performed using Kelly et al study. In that regard, authors should include a local/global sensitivity analysis to evaluate and finalize the model.

Sensitivity analysis was added in the following:

Lines 172-175:

172 A local one-at-a-time sensitivity analysis was performed on the model parameters. We
173 take the -10% and +10% of the baseline values of the parameters, as shown in Table
174 S1. We then gathered the absolute difference of the product (ergosterol and toxic sterol)
175 concentrations from their baseline values, respectively.

Lines 276-280:

276 *3.6 Sensitivity analysis*
277 The local one-at-a-time sensitivity analysis was plotted using a tornado chart for
278 ergosterol, as shown in Figure 4, and for toxic sterol, as shown in Figure 5. The longer
279 the bar a parameter has, the more sensitive the products (ergosterol and toxic sterol)
280 are to it.

Lines 343-347:

343 The local one-at-a-time sensitivity analysis, as shown in Figures 4 and 5, showed that
344 the more upstream the pathway is, the less sensitive the products (ergosterol and toxic
345 sterol) are to them. Whereas the more downstream the pathway is, the more sensitive
346 the products are to them. Also, the products (ergosterol and toxic sterol) are more
347 sensitive to the parameters in the alternative pathway than in the normal pathway.

Figure 4: (lines 542-548)

542 **Figure 4.** Tornado chart of the local one-at-a-time sensitivity analysis of the model
543 parameters to ergosterol. The y-axis enumerates all the model parameters while the x-
544 axis represents the absolute difference of the resulting ergosterol concentration to its
545 baseline value. The red bars indicate the sensitivity analysis of the -10% of the baseline
546 values. Whereas the blue bars indicate the sensitivity analysis of the +10% of the
547 baseline values. The longer the bars are the more sensitive the ergosterol is to them.
548 See Table S1 for the parameters used in the study.

Figure 5: (lines 551-557)

551 **Figure 5.** Tornado chart of the local one-at-a-time sensitivity analysis of the model
552 parameters to the toxic sterol. The y-axis enumerates all the model parameters while
553 the x-axis represents the absolute difference of the resulting toxic sterol concentration to
554 its baseline value. The red bars indicate the sensitivity analysis of the -10% of the
555 baseline values. Whereas the blue bars indicate the sensitivity analysis of the +10% of
556 the baseline values. The longer the bars are the more sensitive the ergosterol is to
557 them. See Table S1 for the parameters used in the study.

Reviewer #2 (Comments for the Author):

Azole resistance continues to be a challenge in treating Candidiasis. Yu et al. developed a mathematical model to predict and understand how resistance is built up by different genes in the pathway. To simplify calculations, ERG11, ERG3, and ERG6 of ergosterol pathway were selected. Sufficient experimental data have been accumulated in the field to understand the role of each gene, which streamlined the validation of the mathematical model. The authors were able to confirm previous experimental findings and offer new insights: while methyltransferase is a good target of antifungals, C5-desaturase is not. Using a computational model circumvents the need to carry out extensive wet lab work to confirm or refute hypothesis.

This work is creative and provides an alternative perspective to understand antifungal resistance. It would be great if the authors could discuss how applicable and generalizable this computational approach is toward other inhibitors on Candidiasis, and inhibition of completely different pathways.

The generalizability of the approach is added to lines 362-363.

362 which is the target of allylamine antifungals. Moreover, the computational approach
363 used in our study is also applicable to other completely different pathways involved in
364 the azole resistance mechanism, such as the *CDR1*, *CDR2*, and *MDR1* genes, which
365 are involved in the drug efflux transporter. We also recommend doing further

Furthermore, to cater to a broader audience, the authors are encouraged to include a biosynthetic pathway of ergosterol and the representative structures of azoles as a figure in the main text. This would make it easier for readers to follow the flow of the paper.

The ergosterol biosynthesis pathway is shown in Figure 2.

Whereas the fluconazole structure is now added as Figure 1.

Reviewer #3 (Comments for the Author):

A. There are a lot of grammar mistakes:

- 1- Line 20 is reported to be not as.
- 2- Line 22 add the word 'The' before most common and delete it before candida in the same line.
- 3- Line 35 add 'an' before adjunct to fluconazole.
- 4- Line 38 replace narrow with narrowing down.
- 5- Line 43 add 'the' before the word most.
- 6- Line 44 remove 'the' before Candida albicans.
- 7- Line 61 remove 'the' before Candida albicans.
- 8- Line 67 add 'the' before the word invasive.
- 9- Line 73 remove the comma after the word epidemiology.
- 10- Line 84 transforms the word non-albicans into singular form 'non-Albican'.
- 11- Line 91 replace in with into before the word azole.
- 12- Line 96 remove the comma after the word experiment.
- 13- Line 112 add 'the' before the word Python.
- 14- Line 118 add 'the' before the word Deletion.
- 15- Line 128 transforms the word "compares" from present tense to past tense.
- 16- Line 134 add 'the' before the word growth.
- 17- Line 200 transform the word case to plural form 'cases'.
- 18- Line 204 removes the comma after the word sterol.
- 19- Line 205 remove 'a' before the word fair.
- 20- Line 208 removes the comma after the word sterol.
- 21- Line 214 transform the word case to plural form 'cases'.
- 22- Line 216 removes the comma after the word albicans.
- 23- Line 218 removes the comma after the word sterol.
- 24- Line 220 add 'an' before the word adjunct.
- 25- Line 223 removes the comma after the word sterol.
- 26- Line 224 remove 'a' before the word fair.
- 27- Line 228 transform the word case to plural form 'cases'.
- 28- Line 230 removes the comma after the word albicans.
- 29- Line 231 remove 'a' before the word fluconazole.
- 30- Line 233 removes the comma after the word sterol.
- 31- Line 234 remove 'a' before the word fair.
- 32- Line 237 removes the comma after the word sterol.
- 33- Line 238 add 'an' before the word adjunct.
- 34- Line 243 transform the word case to plural form 'cases'.
- 35- Line 245 removes the comma after the word albicans.
- 36- Line 247 removes the comma after the word sterol.
- 37- Line 248 remove 'a' before the word fair.

- 38- Line 251 removes the comma after the word sterol.
- 39- Line 253 add 'an' before the word adjunct.
- 40- Line 257 transform the word case to plural form 'cases'.
- 41- Line 259 removes the comma after the word albicans.
- 42- Line 261 and 265 remove the comma after the word sterol.
- 43- Line 266 remove 'a' before the word fair.
- 44- Line 271 add comma after word deletion.
- 45- Line 275 remove 'a' before the word fair.
- 46- Line 279 remove comma after the word content and replace the word accumulation with accumulating.
- 47- Line 281 removes the word genes after ERG6.
- 48- Line 283 remove comma after the word experiment.
- 49- Line 290 remove 'the' after the word C. albicans and 'an' before the word overexpression.
- 50- Line 293 removes the comma after the word sterol.
- 51- Line 303 and 308 remove 'a' before the word fair.
- 52- Line 312 and 314 remove the comma after the word sterol.
- 53- Line 316 replace 'by' with 'of'.
- 54- Line 326 remove 'a' before the word fair.
- 55- Line 340 remove 'doing' before the word costly.
- 56- Line 346 add 'the' before the word azole.

We have accordingly changed all the grammatical corrections described above.

B. Here are some missing details that need to be mentioned:

1- Mention more details about the methods used for the experimental antifungal assay.

We have indicated in the methods section the different assays for antifungal susceptibility on lines 177 - 181.

177 In experimental studies, antifungal susceptibility is quantified by visual spotting assay on
178 YEPD medium, such as in [5], disc diffusion method, such as in [10], or broth
179 microdilution method, such as in [20]. However, these methods are not feasible to use
180 in mathematical models since they cannot be quantified. As such, we look for
181 quantitative methods at the metabolite level to circumvent this limitation.

The assays were not performed in the study since the focus of our research is on the modeling approach.

2- Mention the CLSI version which was used as a reference in the antifungal Assay.

Please see Line 161 for the changes.

160 The clinical breakpoints as determined from the 24-hour Clinical & Laboratory
161 Standards Institute (CLSI) M27-A3 [18] broth microdilution method for fluconazole
162 against *C. albicans* are as follows: Susceptible (minimum inhibitory concentration (MIC)
163 $\leq 2 \mu\text{g/ml}$), Dose-dependent (MIC = $4 \mu\text{g/ml}$), and Resistant (MIC $\geq 8 \mu\text{g/ml}$) [19]. We re-

October 23, 2022

Dr. Angelyn Relucio Lao
De La Salle University
Department of Mathematics and Statistics
2401 Taft Avenue, Malate
Manila, NCR 0922
Philippines

Re: mSystems00691-22R2 (Mathematical Modeling of Fluconazole Resistance in the Ergosterol Pathway of *Candida albicans*)

Dear Dr. Angelyn Relucio Lao:

Your manuscript has been accepted, and I am forwarding it to the ASM Journals Department for publication. For your reference, ASM Journals' address is given below. Before it can be scheduled for publication, your manuscript will be checked by the mSystems production staff to make sure that all elements meet the technical requirements for publication. They will contact you if anything needs to be revised before copyediting and production can begin. Otherwise, you will be notified when your proofs are ready to be viewed.

Publication Fees:

If you would like to submit a potential Featured Image, please email a file and a short legend to mSystems@asmusa.org. Please note that we can only consider images that (i) the authors created or own and (ii) have not been previously published. By submitting, you agree that the image can be used under the same terms as the published article. File requirements: square dimensions (4" x 4"), 300 dpi resolution, RGB colorspace, TIF file format.

We recognize that the video files can become quite large, and so to avoid quality loss ASM suggests sending the video file via <https://www.wetransfer.com/>. When you have a final version of the video and the still ready to share, please send it to mSystems staff at mSystems@asmusa.org.

Sincerely,

Babak Momeni
Editor, mSystems

Journals Department
Fig. S5: Accept
Fig. S3: Accept
Fig. S4: Accept
Fig. S1: Accept
Fig. S2: Accept
Fig. S6: Accept
Table S1: Accept